# Calibrated cryo-cell UV-LA-ICPMS elemental concentrations from NGRIP ice core reveal abrupt, sub-annual variability in dust across the interstadial period GI-21.2

Damiano Della Lunga[1], Wolfgang Müller[1], Sune Olander Rasmussen[2], Anders Svensson[2], Paul Vallelonga[2]

[1]Department of Earth Sciences, Royal Holloway University of London, Egham TW20 0EX, United Kingdom
[2]Centre for Ice and Climate, Niels Bohr Institute, University of Copenhagen, 2100 Copenhagen Ø, Denmark

*Correspondence to*: D. Della Lunga (dellalungadamiano@gmail.com)

**Abstract.**

Several abrupt shifts from periods of extreme cold (Greenland stadials, GS) to relatively warmer conditions (Greenland interstadials, GI) called Dansgaard-Oeschger events are recorded in the Greenland ice cores. Using cryo-cell UV-laser-ablation inductively-coupled-plasma mass spectrometry (UV-LA-ICPMS), we analysed a 2.85 m NGRIP ice core section (2691.50 – 2688.65 m depth, age interval: 84.86 – 85.09 ka b2k, thus covering ~230 years) across the transitions of GI-21.2, a short-lived interstadial prior to interstadial GI-21.1. GI-21.2 is a ~100-year-long period with $\delta^{18}O$ values 3 – 4‰ higher than the following ~200 years of stadial conditions (GS-21.2), which precede the major GI-21.1 warming. We report concentrations of 'major' elements indicative of dust and/or sea salt (Na, Fe, Al, Ca, Mg) at a spatial resolution of ~200 μm, while maintaining detection limits in the low-ppb range, thereby achieving sub-annual time resolution even in deep NGRIP ice. We present an improved external calibration and quantification procedure using a set of five ice standards made from aqueous (international) standard solutions. Our results show that element concentrations decrease drastically (more than tenfold) at the warming onset of GI-21.2 at the scale of a single year, followed by relatively low concentrations characterizing the interstadial part before gradually reaching again typical stadial values.

## Introduction

Dansgaard-Oeschger (D-O) events are abrupt climatic fluctuations between periods of full glacial conditions (called Greenland stadials, GS) and periods of relatively mild conditions during the last glacial (Greenland interstadials, GI) (Rasmussen et al., 2014).

During stadials, deposition of dust and sea salt in Greenland ice significantly increases. Sea salt aerosols in ice cores are present with several species (e.g. $Na^+$, $Cl^-$ and $Mg^{2+}$) as major impurities. The source of these particles is bubble bursting over open ocean water (Lewis and Schwartz, 2004), where winds lash vigorously the sea surface. The aerosols are then transported and deposited on the ice cap. This phenomenon is strongest during stadials but also varies within a year, with the aerosol deposition

peaking in wintertime (Wolff et al., 2003). This is because storminess over the ocean enhances the transport of sea salt species inland during cold conditions, although this effect has to counter the typical increase of sea-ice extent during winter that makes it more difficult for sea-salt aerosols to reach a particular site, since they have to travel further (Petit et al., 1999). This mechanism, which is thought to be the primary reason for sea-salt enrichment in ice cores during cooling events, receives

possibly further contributions of sea salt from another source. When sea ice is formed, highly saline brine and fragile frost flowers form on top of the frozen surface. This brine represents a potential source of aerosol, carried over land by the wind (Wolff et al, 2003). However, from a quantitative point of view, the contribution of brine, frost flowers, and blowing snow to the wintertime peak in sea salt aerosol it is still a matter of debate (Huang and Jaegle, 2016).

Studies suggest that during stadials, the increased storminess and surface wind speed together with the reduced moisture

content in the atmosphere and soil facilitates the sharp increases of continental dust transport to polar areas (Yung et al., 1996; Kreutz, 2013). The source of Greenland dust includes high-elevation sites and high-latitude steppe in Asia whose area increased during cold, more arid periods (Mahowald et al., 1999).

The determination of the phasing of the different records has always been a key aim of high-resolution investigations of Greenland ice cores in order to determine the exact time sequence of variations in temperature, moisture sources, precipitation,

and input of Asian dust and sea salt (e.g. Steffensen et al., 2008). In fact, the phasing of dust records in polar ice cores as inferred from the non-sea-salt fraction of ions (e.g. $Ca^{2+}$, $Mg^{2+}$, $Al^{3+}$, $K^+$, $Fe^{2+}$), which are largely the result of carbonate and silicate mineral weathering (Lewis and Schwartz., 2004), can be used to reconstruct changes of past climatic conditions and atmospheric circulation (Zhang et al., 1997).

Impurities in ice are measured routinely by Continuous Flow Analysis (CFA), which melts a section of the ice core

continuously while measuring different chemical components, such as $Na^+$, $NH_4^+$, dust and conductivity, in the melt water through several detectors. Depending on melt speed and the characteristics of the analytical set-up, layers with thicknesses down to ~10 mm can typically be resolved (Bigler et al., 2011; Vallelonga et al., 2012).

The aim of the present study is to assess the sensitivity and the phasing of dust/sea-salt proxies as $Na^+$, $Fe^{2+}$, $Al^{3+}$, $Ca^{2+}$ and $Mg^{2+}$ at a resolution of ~200 µm (providing approximately 50 data points per calendar year at this depth) across the abrupt

warming into and cooling out of the precursor event GI-21.2. Furthermore, we present an updated fully quantitative calibration for the elements under investigation, following Della Lunga et al. (2014) and Müller et al. (2011).

**Methods and Calibration**

NGRIP ice core samples were initially prepared at the Centre for Ice and Climate, Niels Bohr Institute, Copenhagen. They were cut using a band saw to fit the laser ablation cryo-cell sample holder at Royal Holloway University of London (RHUL),

which is able to simultaneously hold three ice strips of dimensions 50 x 11 x 11 mm (see Della Lunga et al., 2014). For this study a section of 2.85 m of NGRIP ice from the depth interval 2688.65 – 2691.50 m was selected (Fig. 1). This section corresponds to more than two hundred years, given the layer thickness of ~10 mm (Vallelonga et al., 2012). The section covers

an age range of 85.09 – 84.86 ka b2k and therefore includes GI-21.2 (Rasmussen et al., 2014). We utilized samples from a similar position within the ice core cross section as in Della Lunga et al. (2014).

The analytical methodology of cryo-cell UV-LA-ICPMS used for these analyses follows Müller et al. (2009, 2011) and especially Della Lunga et al. (2014) and only a brief summary will be given here. Cleaning of the ice surface has been

conducted using a ceramic 'major-elements free' Y-doped $ZrO_2$ blade (American Cutting Edge, U.S.A.), mounted on a custom-built, acid-cleaned PTFE vice that allows ice scraping in steps of less than 0.5 mm and surface smoothing in order to remove contamination from handling and cutting. Approximately 2 mm of ice were removed from all the surfaces about to be analysed. Handling and smoothing procedures were conducted in a clean hood (US 10-100, ISO4-5) utilizing laboratory gloves.

The adopted methodology includes the acquisition of the following mass/charge ratios: 23(Na), 24(Mg), 27(Al), 34(S), 39(K),

40(Ca), 44(Ca), 55(Mn), 56(Fe), 65(Cu), 85(Rb), 88(Sr), 89(Y), 138(Ba), 139(La), 140(Ce), 141(Pr), 147(Sm), 153(Eu), 157(Gd), 172(Yb), 208(Pb), with dwell times ranging from 5 to 40 ms (see Della Lunga et al., 2014), and a total sweep time of 550 ms. Among these, only the following usually show resolvable signal/background ratio and will be displayed as results: 24(Mg), 27(Al), 40(Ca), 56(Fe). All elements were acquired in *reaction* mode, utilizing 4.5 ml/min of $H_2$ in the octopole cell, allowing the removal of conventional plasma interferences via charge transfer reaction, particularly significant on mass 40(Ca)

and 56(Fe) from $^{40}Ar$ and $^{40}Ar^{16}O$. Formation of hydrides has been monitored on specific isobaric-free masses (210, 233) in no gas and $H_2$ mode and resulted in no significant formation of such compounds in both cases. However, mass 39(K), despite resolvable signal/background ratio, shows a potential interference from $^{38}ArH$ resulting from adding $H_2$ in the reaction cell, and therefore will not be considered further.   Rare Earth Elements were monitored as indicator of further possible contamination due to smoothing and were not the main target of this study.

Intensities of isotopes acquired have been recalculated as elemental intensities based on their relative isotopic abundance (Berglund and Wieser; 2011).

Correction for instrumental drift has been carried out as follows:

$$I_i^{Sa} = I_i^{raw} + \left[t - \left(\frac{t_f-t_0}{2}\right)\right] \frac{1}{k} \sum_{i=1}^{k}[m_{std\_i}]$$    (eq.1)

where $I_i^{Sa}$ is the intensity of element i in the sample and surrounding background corrected for instrumental drift, $I_i^{raw}$ is the

raw intensity of element i in the sample, $t$ indicates the time (in s) of the analysis between the finish time $t_f$ and the start $t_0$ and $m_{std\_i}$ represents the slope of the regression line obtained using NIST612 standard data (Fig S1, supplementary material) acquired for each element during a single ICPMS run executed during a day of analyses where $k$ ICPMS runs are performed. The typical ICPMS instrumental drift observed during a long data acquisition 'run' comprised of standards, cleaning and data acquisition is usually comprised between 5 and 8% per hour, with NIST 612 intensities slightly decreasing with time (see Fig.

S1, supplementary material).

Each element has been externally calibrated using a set of four custom-made ice standards chosen from a total of five (SLRS-5, SLRS-5_10, ICP-20, NIST1648a and Water Low), prepared at RHUL from four different standard solutions at different concentrations and different dilutions (Table S1, see supplementary material). All of our Ice standards except SLRS-5 were

prepared by dilution between 1:10 and 1:1000 of the certified reference material with ultrapure $H_2O$ (>18 MΩ·cm); we very mildly acidified these solutions with 1% ultrapure $HNO_3$ to stabilize them before freezing and to align them with the acidity of the multi-elemental standard solution ICP1 (Sigma-Aldrich), which was the only one being originally (before dilution) in 10% $HNO_3$, unlike all of our other standard solutions..

This external calibration assumes overall comparable ablation characteristics of NGRIP ice and ice standards, which in view of their similar matrix is a satisfactory assumption. Furthermore, using m/z=17 (OH) as an internal standard following Reinhardt et al. (2003), is not feasible because the significantly lower sample consumption of UV-LA relative to IR-LA (Müller et al., 2011) does not result in a background-resolved ICPMS signal at m/z=17. Ice standards were made in a laminar-flow clean hood (US 10-100, ISO4-5) located in a clean laboratory at RHUL, using an acid-cleaned, custom-made PTFE mould

shown in Fig. 2. The mould features two inner volumes, namely one round pool where liquid nitrogen can be used to cool the mould and the innermost volume that uses a polished Pyrex borosilicate glass slide as bottom surface that can be removed to extract the ice. The procedure to produce homogenous ice standards is as follows:

    i.    A polyurethane box is filled with 0.5 l of liquid nitrogen (LN) (Fig. 2.b)

    ii.    1 ml of standard solution already prepared (for concentrations see Table S1, see supplementary material) is

15            pipetted into the inner volume of the mould, to create a ~2 mm liquid layer residing on the glass (Fig 2.a).

    iii.    The entire mould is dipped into the liquid nitrogen, which causes near-instantaneous shock-freezing of the liquid contained in the inner volume (Fig 2.b). The procedure indicated in ii) and iii) is then repeated 5 times to create a volume of ice of ~10 mm height, built up by shock-frozen layers of standard solution.

This procedure ensures acceptable homogeneity of elements in the ice volume at relative standard deviations (RSD) of ~10 –

15 % within a single analysis (Fig. 3), improving on what has been achieved in other UV-LA-ICPMS ice core analyses (Sneed et al., 2015). A standard suspension of NIST1648a has been prepared by carefully weighing 4.92 mg of 'Urban dust' NIST1648 reference material which was subsequently diluted in 100 ml of ultrapure (18.2 MΩ-cm) water and 1 ml of $HNO_3$. The solution then was homogenised through 3 cycles of 5 min of mechanical vibration of the container, before being frozen as described in i) - iii). Given the NIST1648a average particle size of 5 – 10 µm and the 90% percentile of 30 µm, we assume a homogeneous

distribution of particles at the scale of the acquisition spot size utilized (212 µm). Ice blanks were also produced following the procedure described above by shock-freezing ultrapure (18.2 MΩ-cm) water; corresponding UV-LA-ICPMS data show no significant contamination following laser cleaning of the ice surface (see Fig. S.2 and Table S.1 in the supplementary material). For each element, the slope of the equation of the regression line fitting all four standards in a linear plot has been calculated (together with the corresponding $R^2$ value) and utilized to convert net-intensities into concentrations. For the sake of display

Fig. 4 show all regression lines in a log-log plot.

Analyses were carried out using laser tracks which had been preceded by three laser cleaning passages at 25 Hz with a spot size of 280 µm and a speed of 8 mm/min. This was done to remove residual contamination after cleaning with the custom-built vice. Data were acquired at 20 Hz, 212 µm spot size, 3 mm/min speed and a laser fluence of ~3.5 J/cm². This gives a resolution of approximately 200 µm and a cumulative trench depth of ~20 µm (estimated by visual imaging and a typical

ablation rate per pulse of 0.1 µm; Müller et al., 2011). Every acquisition run starts and ends with a NIST612 and ICP-20/SLRS-5/NIST1648a track and comprises two parallel tracks, to assess reproducibility. Fig S.3 (supplementary material) shows raw intensities from two representative parallel ablation tracks running 2 mm apart along the length of three consecutive samples (depth range: 2691.45-2691.30 m). The tracks show that, for each element, the signal preserves its overall shape in both tracks and concentrations show similar absolute values, with differentiations only observable at sub-mm scale, although significant in few cases (Fig. S3). For all the samples, the instrumental-drift-corrected intensities were then averaged between the two tracks and used for calibration. 2D maps of two different 4x4 mm cross sections at specific depths were constructed interpolating the values resuling after calibration from the signal generated by static laser drilling (40s) on a grid of 12 x 12 circular spots of diameter of 128 µm at 200 µm spacing. The intensities obtained from static drilling were corrected as in Della Lunga et al., (2014).

Limit of detection were calculated as follows:

$$LOD_i^{Sa} = \left( \frac{c_i^{std}}{I_i^{std} - I_i^{bkg}} \right) 3\sigma_i^{bkg} \qquad \text{(eq. 2)}$$

where $c_i^{std}$ is the concentration (in ppb) of the element i in the standard, $\sigma_i^{bkg}$ is the standard deviation of the background for an element i, $I_i^{std}$ is the averaged intensity of the element i in the sample and $I_i^{bkg}$ is the averaged intensity of background of element i. The values obtained for this study are listed in Table S1 (see Supplementary material) and range between 0.6 ppb (Ca) and 48 ppb (Na). The Na LOD value is higher due to typical elevated (LA-ICPMS) sodium background, exaggerated by using routinely NIST61x glasses (14±0.1 % m/m $Na_2O$; Jochum et al., 2011) for other LA work. Therefore, Na data present several gaps and are shown here only in overview figure (Fig. 5), mainly to allow comparison with existing CFA-Na data (Vallelonga et al., 2012). Uncertainties have been estimated using the following equation:

$$\sigma_{tot} = \sqrt{\left(\sigma_{nist\_std}\right)^2 + \left(\sigma_{ice\_std}\right)^2 + (\sigma_{id})^2 + \left(\sigma_{ice\_calib}\right)^2} \qquad \text{(eq. 3)}$$

where $\sigma_{nist\_std}$ and $\sigma_{ice\_std}$ represent the relative standard error of the signal acquired during a single run for NIST 612 and the selected ice standard respectively, while $\sigma_{id}$ and $\sigma_{ice\_calib}$ represent the standard errors related to the instrumental drift correction and the calibration and are typical for each element. The total uncertainty $\sigma_{tot}$ is on average about ±16%, $\sigma_{ice\_std}$ contributing with 90% to this value.

**Results**

Results of cryo-cell UV-LA-ICPMS measurements of Na, Mg, Al, Ca, and Fe concentrations across the analysed section of GI-21.2 and GS-21.2 are displayed in Figs. 5-8. For each millimetre of ice analysed, we obtain 40 data points, given the chosen x-y scan speed and the ICPMS sweep time. The resolvable spatial resolution is ~200 µm given the interplay between spot size, stage speed, ICPMS dwell time and laser repetition rate, making down-sampling of individual data points in the form of a moving average necessary. The matching δ¹⁸O profile (Vallelonga et al., 2012) at 50 mm resolution shows a ~4‰ shift to

more positive values between depths of 2691.15 and 2690.70 m, representing the rapid warming into GI-21.2, after which $\delta^{18}O$ gradually returns to pre-warming values (Figs. 1 and 5). The element profiles acquired via cryo-cell-LA-ICPMS show a similar pattern (Fig. 5). The deepest 300 mm of our profile for all of the elements (depth range 2691.50 – 2691.20 m) show relatively high concentrations and several peaks. An abrupt drop is observable around a depth of 2691.20 m, with minor

differences between each element. The variation is very sharp and happens over the space of approximately 10 mm, which, at this depth, represents approximately one year (Fig. 6). Towards shallower depths, most of the elements show, after some characteristic variability, a minimum in concentrations up to a depth of 2690.10 m. At these depths concentrations often fall below LODs, having the lowest values of the entire section. From depth 2690.30 m onwards, $\delta^{18}O$ gently decrease from approximately -37.5‰ to -41‰, representing the cooling phase. In this part, elemental concentrations increase gradually and

the patterns present a higher degree of variability.

Overall, the record can be divided in three main intervals: (i) the deepest 300 mm (2691.50 – 2691.20 m) show relatively high concentrations for every element, with average values of 54, 490, 48, 60, 15 ppb for Na, Ca, Mg, Al and Fe respectively. Around the depth of 2691.20 m an abrupt decrease in all elemental concentrations is observable, with values dropping by a factor of ~10 to average concentrations of 15, 1.3, 1.4 and 1.0 ppb for Ca, Mg, Al and Fe respectively (Na is well below LOD).

The second section (ii) is characterized by low values during the interstadial phase from 2691.20 to 2690.00 m; followed by (iii) a gradual increase in concentrations from depth 2690.00 to depth 2688.65 m, with most of the elements showing recurring short-term variability at multiannual time scales with more than tenfold concentration oscillations.

Figures 6 and 7 show in detail two 200-mm and 300-mm zooms of section (iii). In Fig. 6, we can also observe few minor differences between the respective elemental profiles: at a depth of 2691.28 m a clear peak in Ca, Mg and Al is not mirrored

by Fe; furthermore, Al and Mg drop in concentration before Ca and especially Fe, whose decrease occurs at a shallower depth by approximately 3 to 5 mm. Similarly we observe a peak in Ca, Mg and Al at a depth of 2689.83 m (Fig. 7) that is much less pronounced in the Fe profile, whereas the opposite feature is seen at a depth of 2689.78 m (Fig. 7), where Fe presents a very pronounced peak that is not matched by Al, Mg and Ca. Figure 8 shows a 30-mm zoom comprising 2-3 annual layer peaks identified in both CFA and LA data. LA profiles show the complex structure of a single annual peak to which several minor

peaks contribute. These peaks may reflect single storm events.

Figures 9 and 10 show a collection of maps of calibrated concentrations of the elements under investigation from a 4x4 mm cross section at depth of 2689.78 and 2689.65 m. These sections were chosen specifically from depths were concentrations were high and presented a considerable degree of small scale variability as inferred from our laser ablation profiles.

**LA-ICPMS-CFA data comparison**

For comparison, our cryo-cell LA-ICPMS data have been plotted together in Fig 5-8 with previously published CFA results from the same NGRIP depths (Vallelonga et al., 2012). In contrast to the cryo-cell LA-ICPMS resolution of ~0.2 mm, the CFA profiles of Na, $\delta^{18}O$, CFA-dust and conductivity have a resolution of 3.5, 50, 1.5 and 1.5 mm respectively. The two datasets

show some similarities: between a depth of 2691.50 and 2691.20 m the dust, and partly also the conductivity profiles present relatively high values, similar to what is observed for our elemental proxies, typical of the stadial GS-22 phase. At 2691.20 CFA-dust and LA data are both characterized by a decrease in concentrations, although the LA data show much clearer and abrupt features, marking the start of the GI-21.2 warm phase. Furthermore, minima for the entire section are located between

depths of 2690.95 and 2690.15 m in both datasets. Also, both datasets agree in the shallowest part of the section, showing a more increasing trend starting at 2690.00 m.

In Fig.5, Na data from CFA and LA-ICPMS analyses have been plotted together on the same y-scale. The two datasets show analogous patterns and broadly comparable average values in some of the section, such as between 2690.00 – 2689.25 m (70 ppb and 67 ppb in the CFA and LA-ICPMS profile, respectively). However, LA-ICPMS-Na characteristically is more variable

and differs from CFA data in the intervals 2689.20 – 2688.65 m and 2691.5 – 2691.5 m, where LA-ICPMS Na is either higher or lower relative to CFA-Na, respectively. This, although seems to indicate that there is not an overall systematic shift between the two techniques, highlight the difficulty of LA-ICPMS to detect reliable absolute concentrations for Na, even if the patterns are preserved. This problem could arise from ICPMS-Na background variability, typically very high from day to day, affecting the signal detection and calibration. Moreover, the tendency of Na to show higher concentrations in the proximity of grain

boundaries and junctions complicates more the detection of Na via LA, since laser ablation tracks scan across several boundaries and junctions several times in a single sample, introducing a factor of differentiation that is also reflected in our calibration since it reduces the homogeneity of our ice standards. Therefore, LA-Na profiles still do not agree satisfactorily with CFA-Na and caution must be applied in the interpretation process.

As a further test, we compared the cryo-cell UV-LA-ICPMS data acquired in the frozen state with results from the same three

NGRIP samples analysed via solution-ICPMS after melting (10 ml). The three samples correspond to three different depths in the immediate vicinity of GI-21.2 and representing a wide range of concentrations: early GS-22 (sample 4940A11), late GS-22 (sample 4900A3) and GI-21.1 (sample 4882B4). Results show that calibrated solution data are consistent with our LA-ICPMS data and differ by 5 – 20 %, which is essentially within our margin of error. Sample 4882B4, representing the last part of GS-21.2, shows the lowest concentrations amongst the three samples and also the consistently largest differences between

solution and laser data (see Fig. S4 in the supplementary material).

**Origin of Laser ablation elemental signal**

The intensity of the LA-signal associated to a certain mass/charge ratio, characteristic to one element, is built up by two different contributions: one from soluble ions present in the ice matrix and the other one from dispersed insoluble mineral particles containing the element in their structure. Micro-particles in the NGRIP ice core have a mean grain size between 1

and 2 µm (Ruth et al., 2003) and therefore are too small to be identified unequivocally with our laser camera. Visual inspection of the sample before, after, and during ablation indicated that no residual spatter of the ablation process was deposited back

onto the ice surface after the laser hit the sample, indicating a complete digestion of the material removed by the ablation pulses. This suggests that no fractionation between soluble and insoluble particle is taking place by effect of the laser sampling. We investigated the spatial distribution of Na, Mg, Al, Ca and Fe over two small horizontal planes (i.e, perpendicular to the core length axis) by analysing 2D maps of concentrations across two specific cross sections (Fig 9 and 10). These sections were constructed interpolating several acquisition points obtained via static laser drilling. Fig 9 and 10 both show concentrations spanning over a range of several tens of ppb for each element across the entire sections. The cross-sections intersect few grain boundaries and junctions (as observable in the laser camera image). The grain boundary net has been overlaid in black onto the elemental maps and shows that, in most of the cases, high concentrations areas are located in the proximity of boundaries and junction, broadly mimicking their pattern. In both cases, these patterns are somehow clearer for element like Na and Mg, related to sea salt, and become less defined going from Ca to Al and Fe. This might be associated with the fact that the elemental signal has a relative increasing contribution from micro-particles going from Ca to Al, to Fe, whereas the contribution from micro-particles to the Na and Mg signal is minimal. This would also suggest that micro-particles are slightly less inclined to be aligned on boundaries and junctions compared to soluble impurities and therefore generate a less defined pattern of concentrations in our maps.

**Discussion**

Our fully quantitative calibration of cryo-cell UV-LA-ICPMS net count rates to elemental concentrations is presented here for the first time. We have succeeded in producing suitably homogeneous ice standards (±10–15% RSD, Fig. 3) from four different solutions at known elemental concentrations and one frozen suspension at different dilutions. This represents an improvement to what has previously been achieved in ice standard preparation (Reinhardt et al., 2003, Wilhelm-Dick, 2008; Sneed et al., 2015). The correlation between the elemental concentrations in the standards and the resulting net-signals from cryocell-LA-ICPMS (in counts per second, cps) is good and follows the expected linear relationship (Fig. 4), with $R^2$ values ranging from 0.89 and 0.98.

The removal of contamination is ensured not only by surface-smoothing executed via a 'major-element free' $ZrO_2$ blade (Della Lunga et al., 2014), but also by laser cleaning performed three times before each acquisition. Its effectiveness can be demonstrated using ice blanks (see Fig. S2 in the supplementary material). The overall uncertainties estimation derived from analysis and calibration gives an average value of ±16%, which has to be considered acceptable for ice core analysis where elemental concentrations are typically in the low ppb range and variability usually covers more than one order of magnitude. Fig. 5 shows remarkably large concentration variations of all the elements, which can drop and rise by a factor of ~10 in as short as 10 mm, representing approximately one year at this depth and confirming that dust proxies (Na, Mg, Al, Ca, Fe) do react to natural abrupt climate change events at a time scales much shorter than the duration of short-lived interstadials such as GI-21.2. The pattern of all the elements shows high values in the deepest part before abruptly decreasing approximately by a factor of 10 down to few ppb or even ppt (below LOD). Concentrations stay low during the GI-21.2 interstadial part and then

rise again more gradually showing much more pronounced oscillations, with a further increase to higher values towards the end of the section, where concentrations return to the typical high stadial concentrations. Overall, the general pattern of LA-ICPMS proxies agrees well (Fig. 5, 6 and 7) with the previously published dataset of CFA analysis for the same NGRIP depth range (Vallelonga et al., 2012).

The slightly different pattern between $\delta^{18}O$ and elemental proxies has to be expected as the resolution of the two records is different, namely 50 mm and ~200 µm respectively. However, LA data seem to confirm that elemental 'dust' proxies react before $\delta^{18}O$ to the GI-21.2 warming onset, showing a drop in concentration at 2691.20 m, thus 100 mm before the main oxygen rise at 2691.10 m, extending and confirming the observations by Thomas et al. (2009).

CFA analysis on the same section show similar features to what we observe in UV-LA data, especially regarding the transitions
from GS-22 to GI-21.2 and from GI-21.2 to GS-21.2, which occur approximately within the same depth range in both cases (2691.20 m, 2690.10 – 2689.90 m). However, elemental proxies (Fig. 5) show much clearer features in terms of abruptness and amplitude of oscillations compared to CFA data, and a more pronounced variability at the cm-scale (Fig 6, 7) that is often related to sub-annual variations, observable also in Fig. 8. This may be related to single storm events that could have originated from different dust sources, resulting in a variation in the elemental ratios (especially Ca-Al vs. Fe) at short-time scales, as
observed in Fig. 6 & 7.

Most of the differences between CFA and LA-ICPMS proxies are observed at a small scale and are mainly influenced by few factors, the first of which is the effect of sample volume. In fact, we estimate that every LA-ICPMS data point corresponds to ~120 ng of ablated ice (based on scanning speed and ice crater depth) whereas CFA sampling resolution is about 0.1-1 g for each data point (Vallelonga et al., 2012). This introduces a difference in the sampling volume between the two datasets that
can also be influenced by surface effects and especially by the wavy nature of layers at this scale and core depth. This is particularly important for Na, whose lateral variability induced by any non-horizontal layering is also affected by diffusion of Na that has been observed at this depth, resulting in a smoothing of the CFA annual signal (Vallelonga et al., 2012). Furthermore, the CFA insoluble dust data presented here refer to measurements of particles of size >1 µm and therefore do not account for insoluble impurities of sub-micron size (Vallelonga et al., 2012).

The elemental maps shown in Fig. 9 and 10 demonstrate that, at sub-cm scale, the concentrations of impurities is strongly influenced by the presence of boundaries and junctions even when considering horizontal planes, whose original impurity-input is assumed to be roughly identical. This introduces a main source of differentiation between LA and CFA sampling and can account for some of the small-scale variability we observe in the LA-profiles. This is again particularly relevant for element like Na and Mg whose 2D distribution seems to follow closely the grain boundary net, presenting higher concentrations in the
proximity of boundary and junction. On the other hand, 'dust'-proxies as Ca, Al and Fe, do not show such a closer overlap of high intensity and presence of boundaries or junctions, possibly as a result of being increasingly associated with insoluble micro-particles dispersed in the ice matrix, which indeed constitutes the CFA-Dust signal. This would suggest that micro-particles in the ice matrix are less inclined to reside on boundaries and junction compared to soluble ions and is consistent with

previous studies of deep ice cores (Della Lunga et al., 2014; Eichler et al., 2016). As a result, the averaging of LA-signal between two or more parallel tracks spaced by few mm is not only desirable but necessary.

Our LA-ICPMS data suggest that dust and sea salt proxies undergo extremely abrupt, namely sub-annual, variations during abrupt climatic change, representing most of the drop/rise in phase with CFA data from the same depth range (Fig. 5, 6 & 7).
As previously observed by Steffensen et al. (2008) and Fuhrer et al. (1999) in the NGRIP and GRIP record for the much shallower GI-1 and GI-3 respectively, the variations of insoluble dust and $Ca^{2+}$ concentrations can occur abruptly at a yearly scale for warming transitions. In contrast, for the cooling phase, the interstadial to stadial switch takes place more slowly and through several oscillations. This is compatible with the cryocell-LA-ICPMS data observed in Fig. 5, extending these patterns to one of the oldest and shortest interstadial-stadial transitions in the NGRIP record. Any mechanism responsible for these
changes must be capable of producing a series of extremely abrupt shifts, and must be able to switch on and off very quickly. A plausible explanation for short precursor-type events such as GI-21.2 could arise from a reorganization of atmospheric circulation at mid-high latitudes in the Northern Hemisphere. This enhances the mobilization at the dust sources (i.e., Asian deserts), as proposed by Fuhrer et al. (1999), and increases the residence time of particles in the atmosphere, which can account for most of the changes in concentration of proxies observable for GI-21.2. GCM simulations (Kutzbach et al., 1993) showed
that during the LGM, storms strengthen their intensity and changed their trajectory originating further south and changing the pressure regime over central Asia. Even a very small increase in the maximum wind speed during episodic storms could have overtaken the threshold value for mobilization of particles of a certain size (Gillette and Passi, 1988). The first signs of the rapid warming could therefore be coeval with a decrease in Ca, Al, Mg and Fe concentrations as a result of wetter conditions in the Asian dust-source areas, where dust uplift was reduced by the increasing humidity and washout following an
intensification of precipitation. A rapid change in atmospheric transport patterns and the relative variation in dust sources would also explain sporadic changes in elemental ratios (e.g., Fe/Ca, Fe/Al), which can be identified in our profiles (Fig. 6 & 7).

## Summary and Conclusions

Using cryo-cell UV-LA-ICPMS we obtained 2.85 m of dust profiles (Na, Mg, Ca, Fe, Al) from 85 ka-b2k-old NGRIP ice
covering the GS-22 – GI-21.2 – GS-21.2 transitions at a resolution of ~200 µm, which corresponds roughly to 50 data points per year. Quantification of LA-ICPMS signals was possible using a set of five external ice standards carefully produced at RHUL, which proved to be homogeneous at the ~15% level. Our results for the short-lived GS-22 – GI-21.2 – GS-21.2 transition show that dust proxies vary by up to ~tenfold in concentration at a scale of ~1 year, showing abrupt drops due to rapid warming also in the deepest (and oldest) part of NGRIP record, similarly to what previously observed for GI-3 and GI-
1 (Fuhrer et al., 1999; Steffensen et al., 2008). During the rise that corresponds to the cooling transition, concentrations do not vary sharply, but gradually following an increasing trend characterized by more than one oscillation. The comparison of cryo-cell-LA-ICPMS profiles with CFA data of Na, dust and conductivity corroborates the results, showing that cryo-cell-LA-

ICPMS profiles present more variability and a larger frequency of high-concentration peaks across the entire record. We suggest that wetter conditions at Asian sources could have abruptly lowered dust uplift and increased the washout during the GI-21.2, when atmospheric circulation over Asian deserts was weaker. This would have resulted in a reduction of transport efficiency and therefore a rapid decrease in dust available to Greenland at short time scales. At the onset of the following cooling period, the end of the wet conditions together with an increase in wind speed and storminess above a threshold level allowed uplift of more particles, which explains the subsequent rise of concentrations of dust to pre-warming levels.

**Author contribution**

DDL designed the experiment, performed the analysis, interpreted the data and wrote the manuscript. WM helped designing the experiment, performing the analysis and the data interpretation and edited the manuscript. SOR and AS contributed to the designing of the experiment, the sample preparation, the data interpretation and edited the manuscript. PV provided CFA data for comparison, helped with the data interpretation and edited the manuscript.

**Acknowledgements**

This work has been supported by a RHUL studentship granted to Damiano Della Lunga, with the analytical costs being co-funded initially via a research grant from Resonetics LLC & Laurin Technic to Wolfgang Müller, and subsequently via a Postdoctoral grant from Australian Scientific Instruments (ASI) to both Damiano Della Lunga and Wolfgang Müller. The authors would like to thank Jerry Morris for continuing invaluable technical support at RHUL. Initial discussions with Michael Kriews and Dorothee Wilhelms-Dick helped to improve the methodology of ice standard preparation.

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

**Figures Captions**

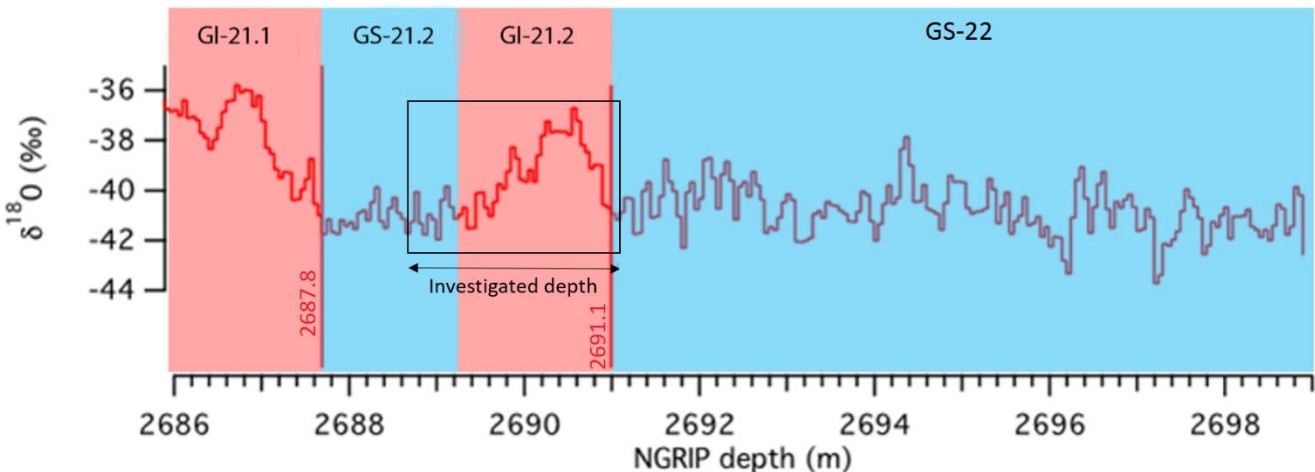

**Figure 1: δ¹⁸O profile across the transition from GS-22 to GI-21.1 (modified from Vallelonga et al., 2012). Stadial and interstadial periods are highlighted in blue and red, respectively. The black box and arrow indicate the corresponding section of ice core analysed for this study.**

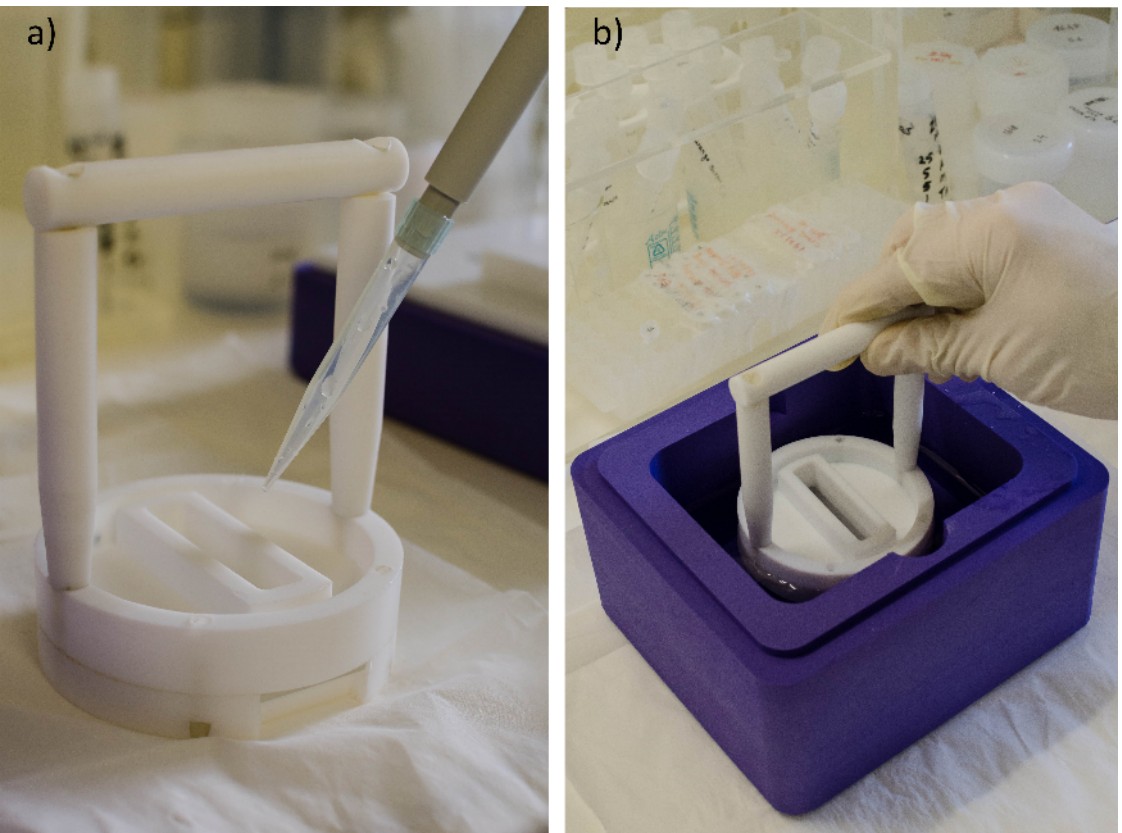

**Figure 2: Ice standard preparation at RHUL. a) 1 ml of aqueous standard solution is pipetted into the inner volume of a PTFE mould featuring a removable glass surface at the bottom to allow the solution to spread uniformly creating a thin layer of water. b) The mould is dipped into liquid nitrogen to instantaneously shock-freeze the solution. This procedure is repeated five times to build up an ice volume by shock-freezing layer by layer of 5 ml total volume resulting in an ice volume approximately 45x10x10 mm. Each ice standard was then surface-cleaned using our PTFE vice before analysis (see text).**

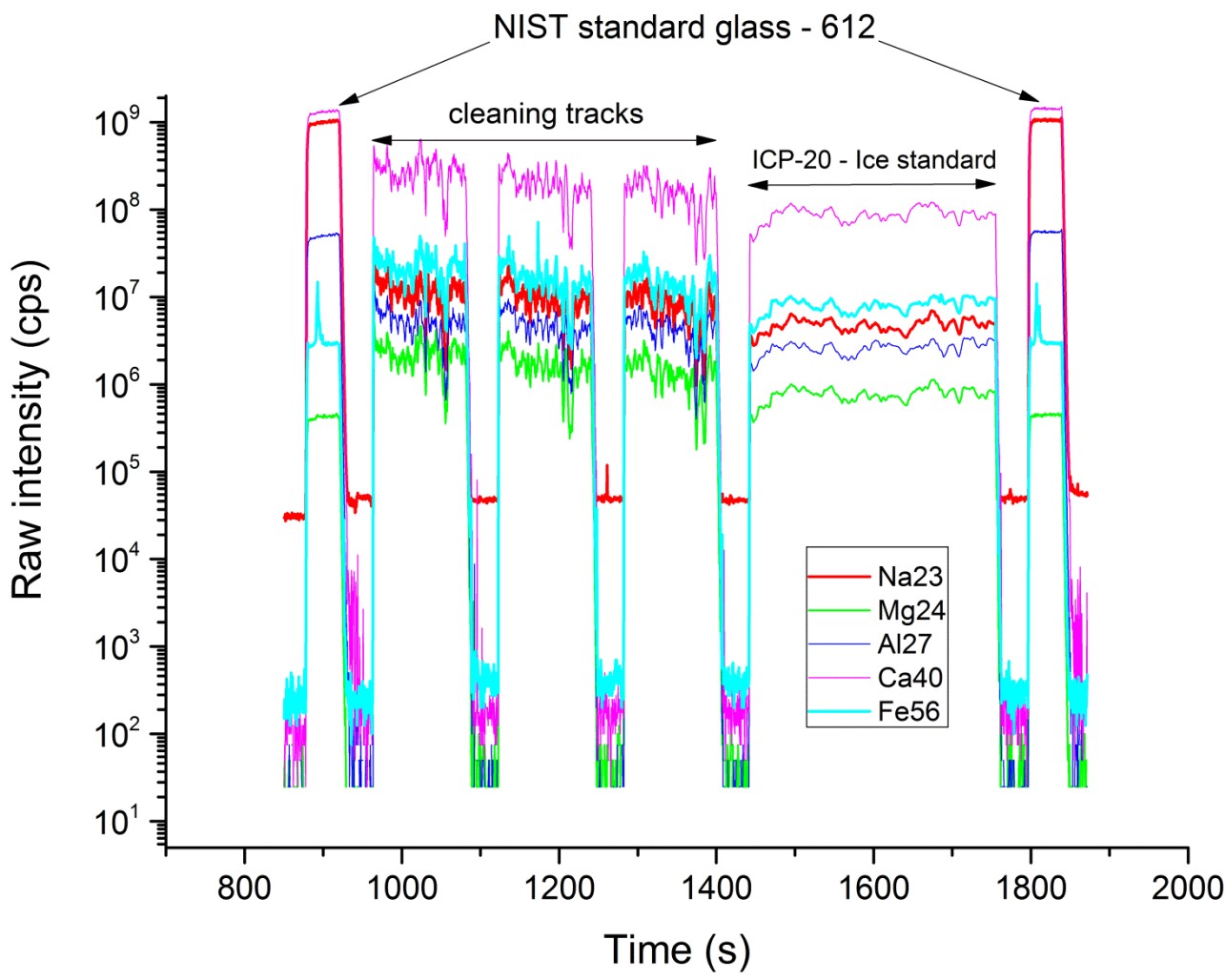

**Figure 3: Example of raw intensity data of NIST612 glass (first and last peak) compared to one of the ice standards prepared for this study (ICP-20). Standard data were acquired following three cleaning runs, and show that the ice standard appears rather homogeneous with typical RSD values between ± 10 and 15 %. See text for details.**

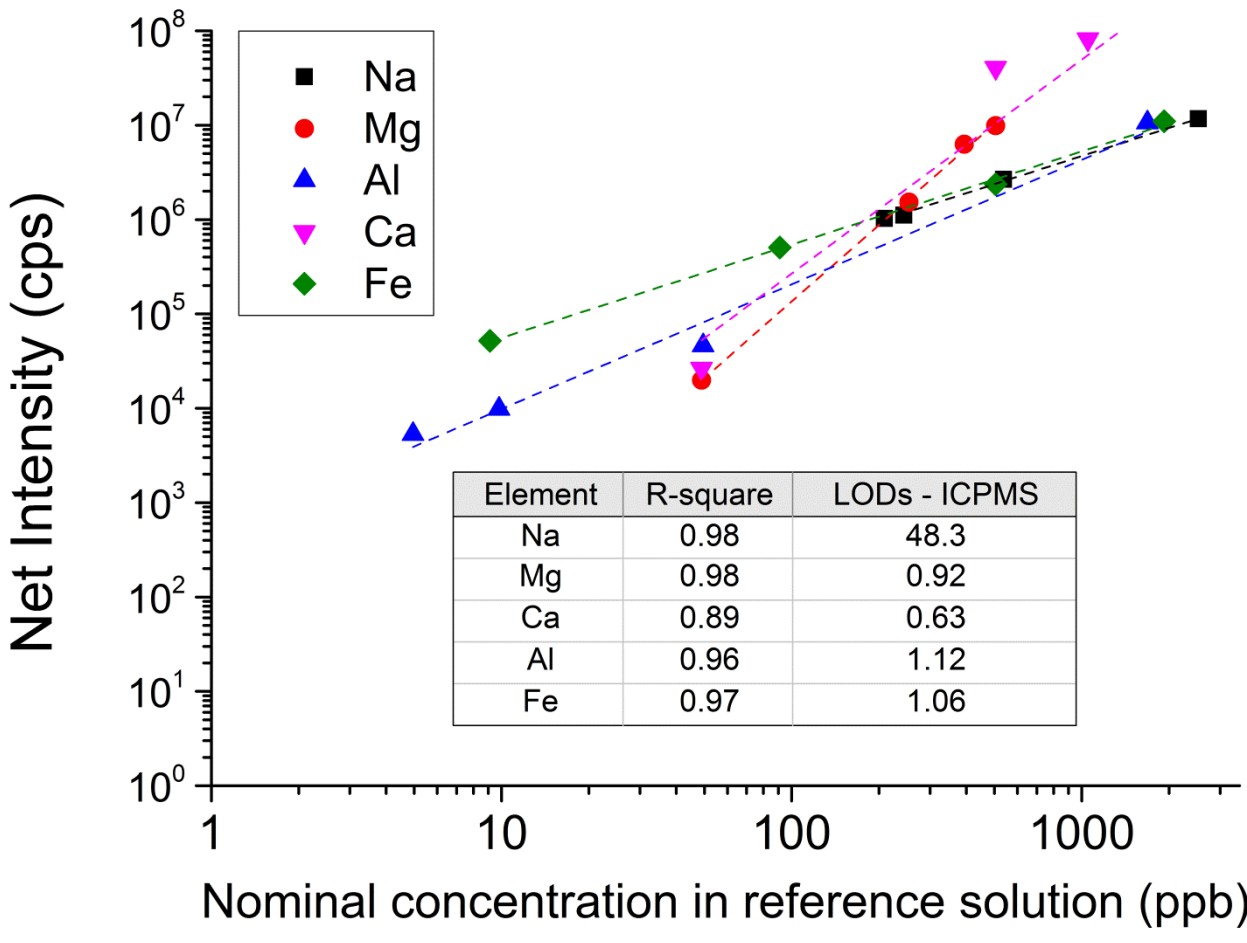

**Figure 4: Calibration lines for elements under investigation obtained utilizing the ice standards listed in Table S1 (supplementary material). LOD indicates limits of detection. See text for details.**

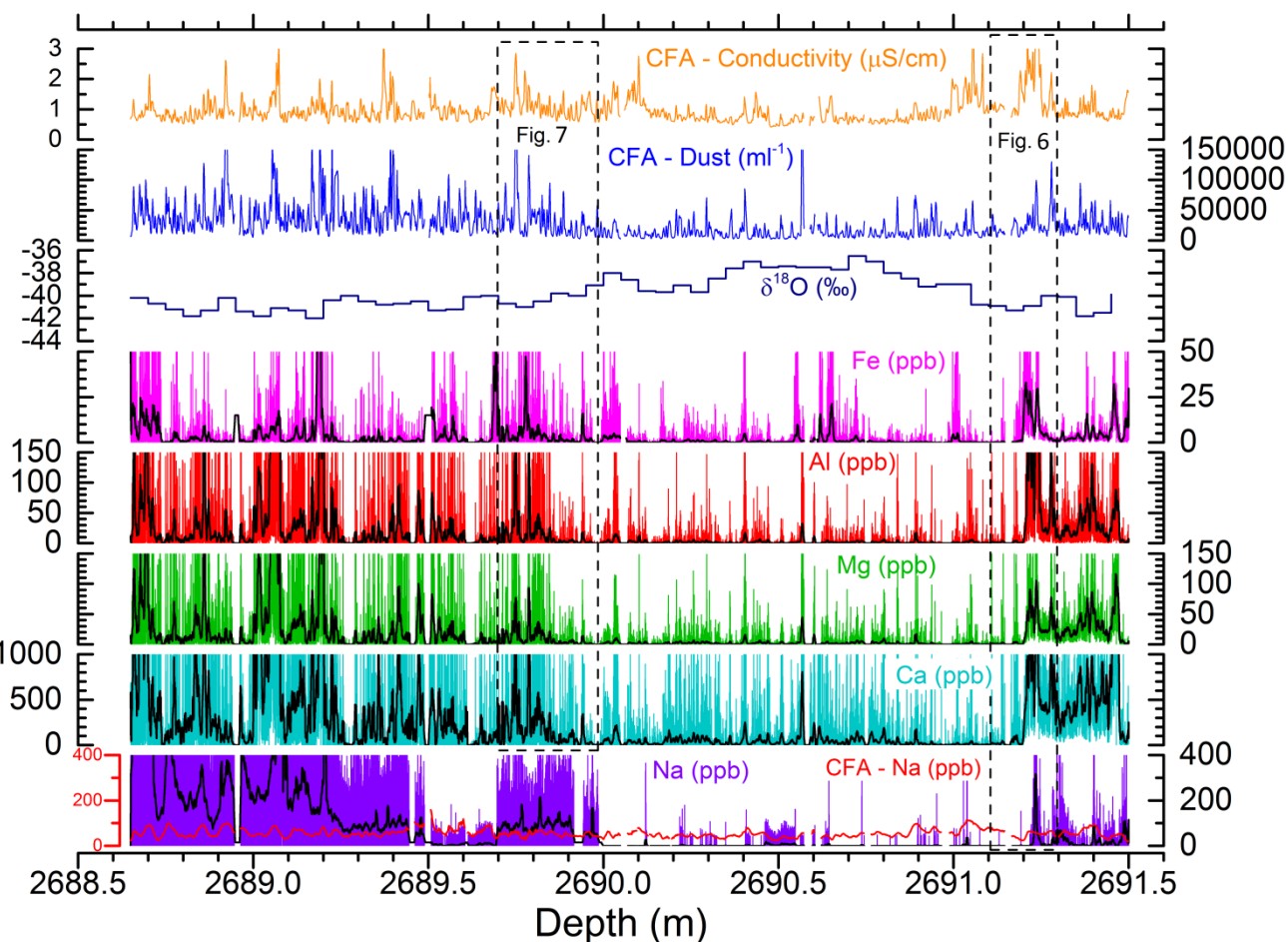

**Figure 5:** Cryo-cell-LA-ICPMS element concentration profiles of Na, Mg, Al, Ca, and Fe and corresponding Na, $\delta^{18}O$ and CFA-dust profiles at 3.5, 50 and 1.5 mm resolution respectively (the latter three from Vallelonga et al., 2012) across 2.85 m of NGRIP ice core that spans from approximately 85090 to 84860 a b2k (±20 a) and contains GI-21.2. The coloured lines are individual LA-ICPMS data points; black lines represent adjacent-element moving average (period 200). It should be noted that cryo-cell-LA-ICPMS Na LOD is 48.3 ppb, which renders most of the interstadial and some stadial Na data undetectable. Overall, Na is mainly shown to allow some comparability with existing CFA Na data (Vallelonga et al., 2012). See text for details.

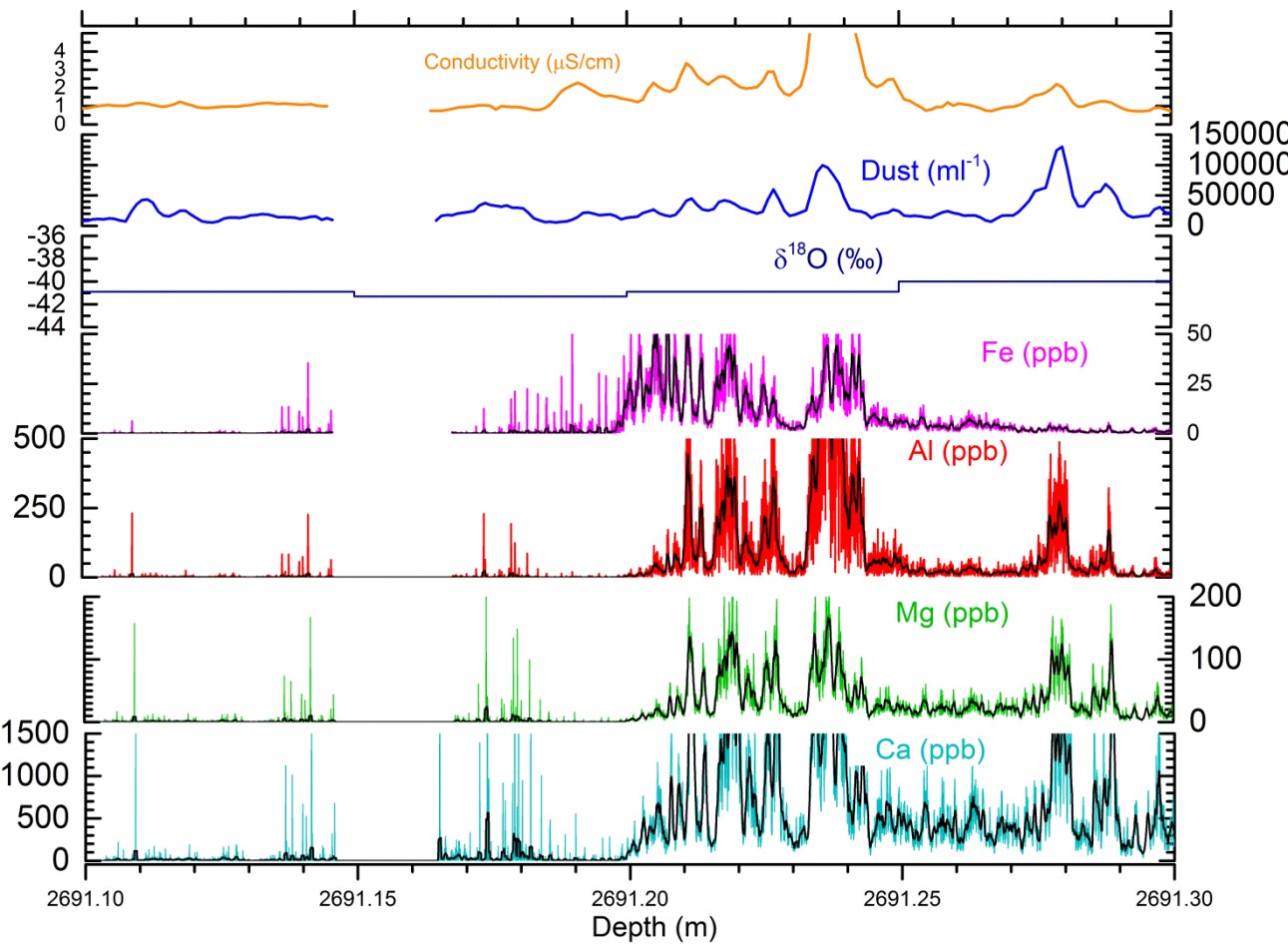

**Figure 6: Zoomed-in cryo-cell LA-ICPMS profiles of a 200 mm window from the deepest part of the GI-21.2 section (cold/warm transition), analysed for the most significant elements and spanning about two decades around 85.1 ka b2k. Coloured lines represent LA data, black lines are 30-points moving averages. A switch between stadial and interstadial typical concentrations is observable around 2691.20 m, happening over the space of just ~10 mm. Conductivity and CFA-dust are from Vallelonga et al. (2012).**

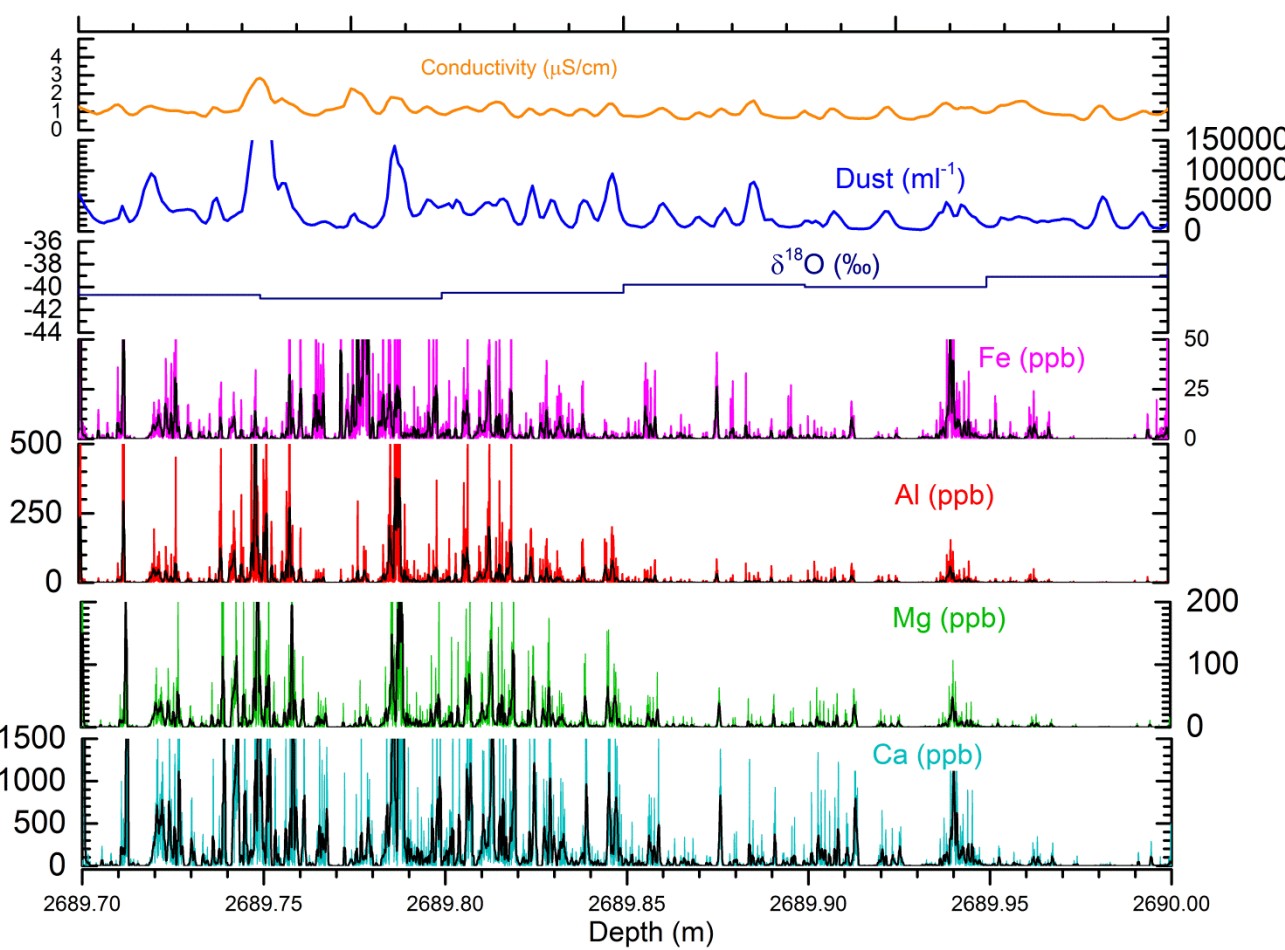

**Figure 7: Zoomed-in cryo-cell LA-ICPMS profiles of a 300 mm window from the middle part of the GI-21.2 section analysed for the most significant elements and spanning about two decades around 85.0 ka b2k (cold-warm transition). Coloured lines represent LA data, black lines are 30-points moving averages. A gradual increase in dustiness is observable starting from a depth of 2689.95 m going towards shallower depths, representing the GI-21.2 – GS-21.2 transition, which in this case takes place over the space of ~150 mm. Conductivity and CFA-dust are from Vallelonga et al. (2012).**

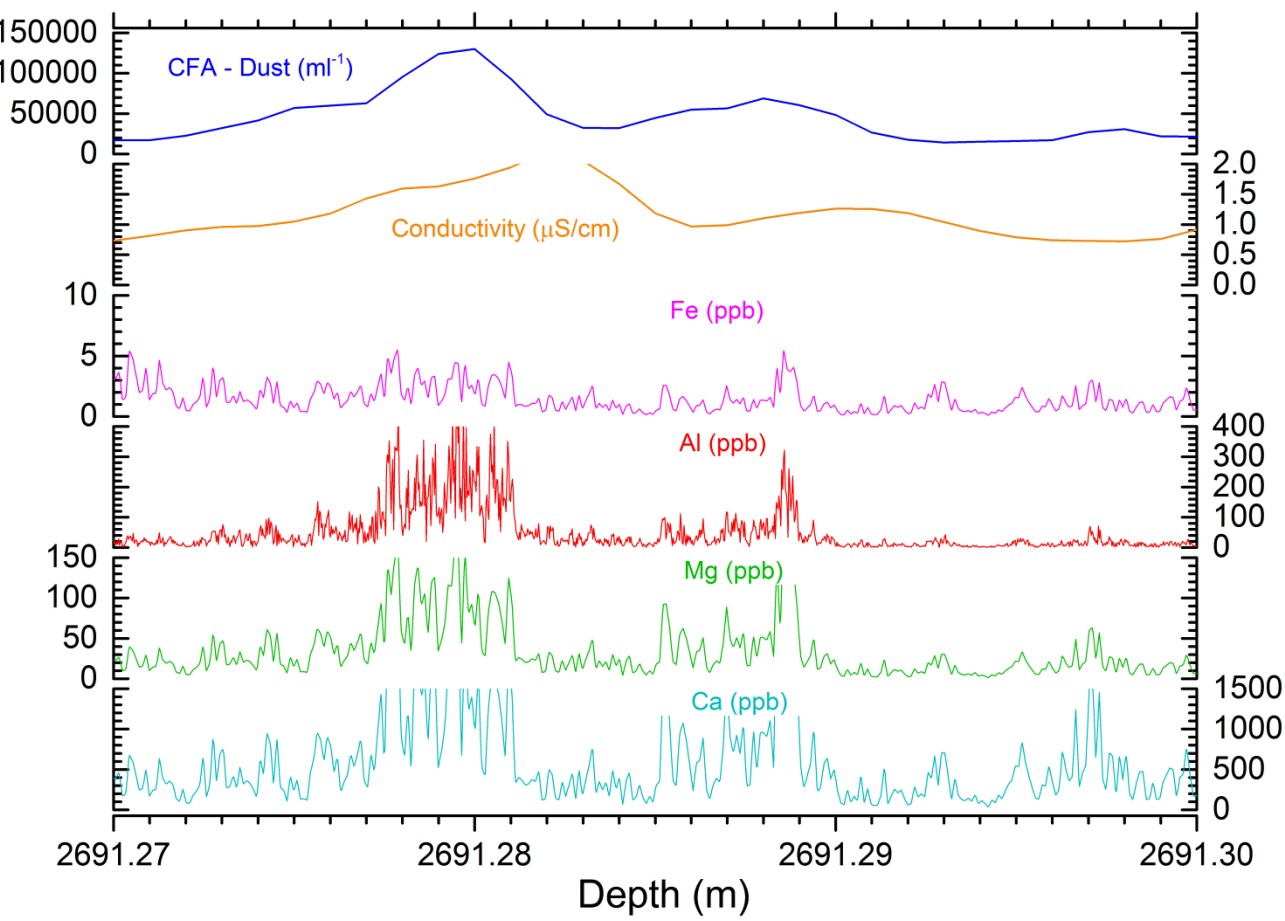

**Figure 8: CFA conductivity, CFA dust, LA-Fe, LA-Ca, LA-Al and LA-Mg direct comparison across a detailed 3-cm zoom. In this case, laser ablation data have not been smoothed. Conductivity and CFA-dust are from Vallelonga et al. (2012). The profiles show sub-annual variations that contribute to the CFA annual signal.**

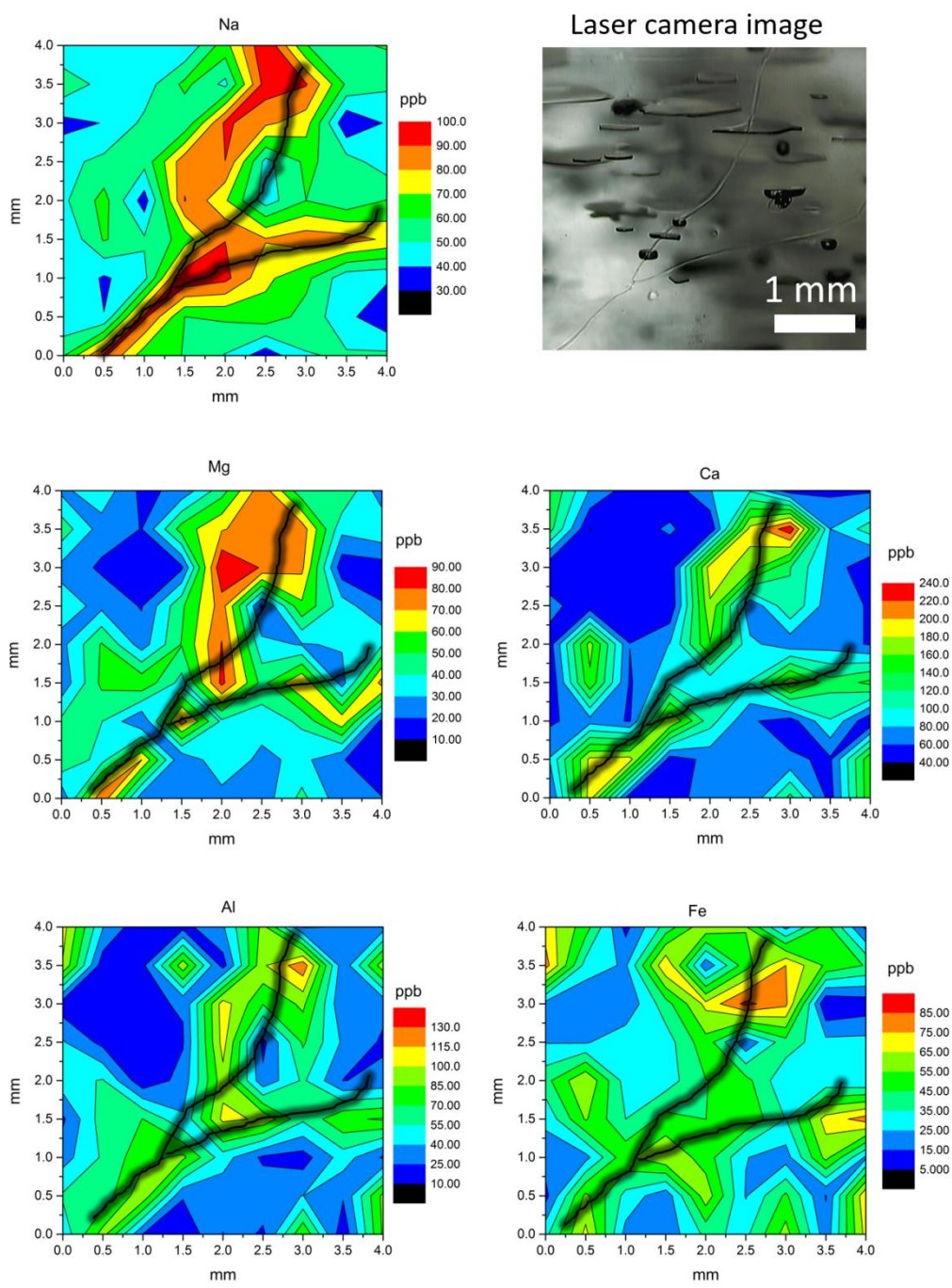

**Figure 9: 2D maps of calibrated concentrations of elements under investigation (Na, Mg, Ca, Al, Fe) across a 4x4 mm cross section with overlaid grain boundary net in black as observed in transmitted light (upper right ) from a depth of 2689.78 m.**

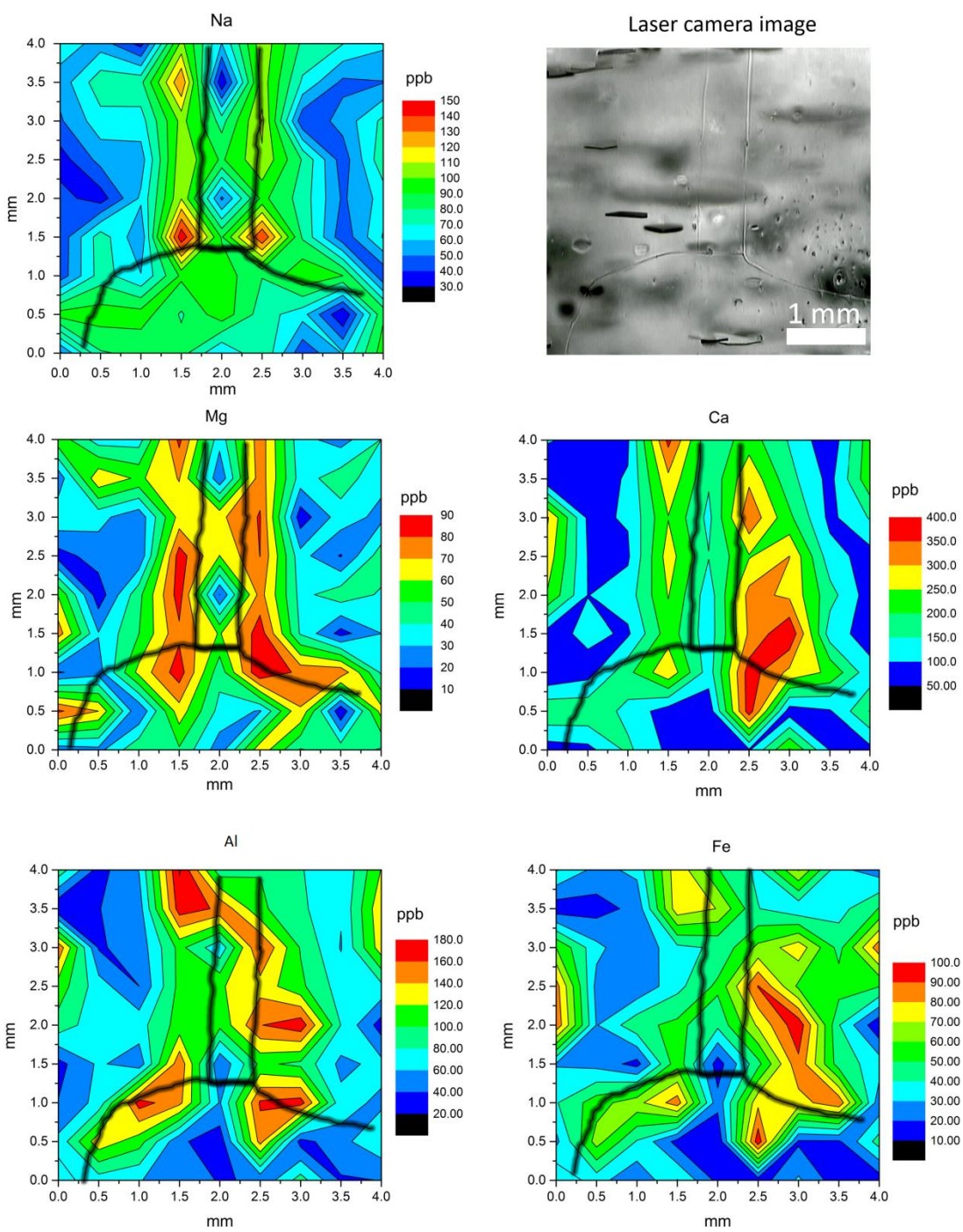

**Figure 10: 2D maps of calibrated concentrations of elements under investigation (Na, Mg, Ca, Al, Fe) across a 4x4 mm cross section with overlaid grain boundary net in black as observed in transmitted light (upper right ) from a depth of 2689.65 m.**