# Peer review of "Calibrated cryo-cell UV-LA-ICPMS elemental concentrations from NGRIP ice core reveal abrupt, sub-annual variability in dust across the interstadial period GI-21.2"

_The Cryosphere, 2016_

## Referee Comment (RC1) · Anonymous Referee #1 · 20 Dec 2016

This paper presents the application of a new high-resolution (~200 µm) analytical technique to a period of abrupt climate change in the NGRIP ice core (actually a 2.85 m section representing about 250 years). I actually find this paper quite hard to review: on the one hand the technical achievement is good and worth documenting. On the other hand what we learn from it is minimal, and there are many more interesting things the authors could have done. I therefore think the authors have two choices. One alternative is that they should shorten the paper and just present it as a proof of concept. The other is that they should add to it – possibly involving new analyses but certainly new data treatments, to try to give new insights into what benefits such a technique might bring.

The positive part is that the authors have successfully used laser ablation to determine 5 elements at 200 µm resolution. They describe the way they cleaned the samples (partly with the laser) and the novel way in which they produced quasi-homogeneous standards. I congratulate them on this.

The headline findings from the study are not new: that dust elements change very rapidly (annual scale) at the start of a D-O event (this was already said as far back as Fuhrer et al 1999), and that they appear to change before the water isotopes (already covered by Steffensen et al and Thomas et al). Sure, this is the oldest section on which such a finding has been confirmed, but as only one event is studied it just adds an example rather than offering a generalisation, and certainly doesn't provide evidence to make new ideas about the mechanism. Of course, this is not the authors' fault. On the other hand they could have taken the opportunity to really discuss what the advantages and drawbacks of such high resolution might be. I can suggest several lines of study they could have taken:

1. An obvious issue is how reproducible the data from such narrow tracks are. The authors say they ran parallel tracks but then do not show us the data so we can assess. I don't know how far apart the tracks were, but parallel tracks across the core at cm distances would have given a crucial clue to reproducibility, which in turn would allow a conclusion as to whether the advantages of high resolution are real (providing evidence of climate variability) or illusory (providing evidence of depositional noise).
2. A second issue concerns diffusion. It is generally assumed that water isotopes diffuse a few cm in the firn and then also in solid ice, sulfate peaks appear to diffuse, while dust probably does not diffuse. What about these elements? Here are data apparently showing the retention of mm scale structure at 80 ka ago. This is interesting in its own right and would be even more so if compared to the structure at the start of DO events in the younger part of the record. It might even have been possible to derive diffusion coefficients, which might be crucial when investigating even older ice (eg in Antarctica).
3. What is this method actually analysing and how does that compare to what CFA and IC measure? We are shown a comparison only for Na (not counting dust which cannot be compared quantitatively). Why? This seems crucial and even if the data are not yet available from the CFA for eg Ca (which is odd if Na has been measured), it would have been trivial to prepare a few 1 cm samples for IC analysis. This seems critical because Fig 5 seems to show unexpectedly poor agreement for Na, which certainly needs discussion. But in general the consideration of whether this method measures more of the insoluble component than CFA/IC would have been an important analytical discussion that could have been included.

I will discuss a few details below, but as already outlined, the issues above could be discussed; if the authors prefer not to then the paper should be cut back to an analytical proof of concept.

Detailed comments:

Page 2, para 1. You seem to come down on one side of an ongoing discussion about whether the cold period enhancement is mainly due to increased transport or to the presence of a sea ice source. It would better reflect the science if you left that open.

Page 2 line 23. It gives a misleading impression to state that rge resolution is "nominally…weekly" because precipitation intermittency and snowdrift mean that weekly resolution is certainly not available. I suspect you know that with your use of the word "nominally", and you should explain that.

Page 3, line 1. You say that the section "covers" GI21.2, and then give an age range of 370 year (84.70-85.07 ka) for that. But in the abstract you refer to it as a 250 year section, even though Fig 1 shows that it is actually wider than GI21.2. This is incompatible – please correct.

Page 3, line 16. Sorry to be picky but you cite Fig 5 before Figs 2-4.

Eq 1 and line 25 is confusing. If I understand it $m_i$ is the slope of intensity vs time, whereas your wording made me think it was the slope of the calibration (intensity vs standard concentration). Please clarify. I think Fig 1 would be better shown as linear rather than log plots, as the log plot hides the extent of the drift.

Page 4, line 25. Is this the $R^2$ of lin-lin or log-log plots? You show log plots but then describe it as a linear regression. Please clarify.

Page 4, line 31. Here is where you say you analysed two parallel tracks to assess reproducibility but then you never do so.

Results, page 5-6, seems repetitive (last para page 5 and first para page 6). Combine them into something clearer?

Page 6, line 9, should be Figs 6 and 7 not 8 and 9.

Page 6, data comparison, lines 27-32. It is clearly not true that Na is comparable between the two techniques. While they match OK at 2689.7-2690.0, they are at least a factor 3 off in the shallower section. This needs a better and more correct discussion. (And of course I would like to see the same for Ca).

---

## Referee Comment (RC2) · Anonymous Referee #2 · 5 Feb 2017

This is an interesting MS showing the powerful application of LA-ICPMS for high resolution (200 um) ice core analysis. This may have a lot of implication especially for low accumulation sites and/or abrut changes.

There are some novelties in this paper on the way the authors prepared the standards to convert the count per seconds (intensities) into concentrations. However they use standard riverine waters (SLRS) and a suspension of NIST648 leached with ultrapure HNO3, which resulted in an ice matrix standard which is far from the real ice matrix. We know that the slopes of the calibration curves are highly dependent from the matrix itself

and I think therefore that the results may be strongly biased by the different ionization of the ice matrix compared to the standard ones. The authors should comment on this and clearly demonstrate how the obtain real concentration and not just relative changes that can be easily seen just looking at the variation of intensities.

In addition, the authors claim to use 24Mg, 27Al, 40Ca, 56Fe. All these masses are highly interfered by spectral and matrix interferences in ICPMS. Despite I think they used a SF-ICP-MS or a collision-cell instrument to reduce the interferences, I think that a better description of the methodology should be given. I know that most of the details are given in Della Lunga (2014), but a minimal description of the methodology is compulsory.

Then, I do not fully understand the objective of the MS, since most of the finding are not new at all. I would have rather focussed the MS into a comparison between LA-ICPMS vs CFA, but this would have required a more robust statistical tool.

I would therefore suggest the authors to readdress the MS to a specific target: i) analytical (in this case the paper lacks of many details), describing in detail the new advancement of this powerful technique and duly comparing the data with CFA results; in this case the reproducibility of the analysis on different sections is a key parameter, but as far as I can see there are no evidence of this in the paper; ii) more oriented toward a climatic/environmental interpretation; in this case the real benefit of the LA-ICP-MS approach should have been better explained.

---

## Author Comment (AC1) · 2 Mar 2017

The Authors would like to thank anonymous reviewer R1 for the very helpful comments, following which we have decided to revise the manuscript in order to include more figures and discussions concerning the small-scale variability and the soluble/insoluble origin of the LA elemental signatures observed. Our replies are here in bold. Relevant changes in text and figures are illustrated at the end of this document, where added or modified text is highlighted in yellow.

1) This paper presents the application of a new high-resolution (~200  $\mu$ m) analytical technique to a period of abrupt climate change in the NGRIP ice core (actually a 2.85 m section representing about 250 years). I actually find this paper quite hard to review: on the one hand the technical achievement is good and worth documenting. On the other hand what we learn from it is minimal, and there are many more interesting things the authors could have done.

It is correct that our paper focuses on the methodology and novel calibration with one casestudy, which primarily aims at highlighting the now achievable sub-annual resolution in very deep ice cores via cryo-cell LA-ICPMS and which is not possible conventionally. On the other hand, our data do provide further evidence for an extremely abrupt mechanism that sustainably changes 'dust' proxy concentrations across Stadial-Interstadial transitions, and which – as R1 also points out – we crucially extend to this ice core depth and for such a shortlived DO event (GI.21.2 – also defined 'precursor' event after Capron et al., 2010, where conventional CFA hardly resolves such rapid changes.) Extending this to one of the earliest DO-events in conjunction with what had been observed in other DO events (Steffensen et al., 2008; Thomas et al., 2008), will eventually allow the community to edge closer to identifying a mechanism driving these changes. To study especially the early Stadial-Interstadial events, we provide both the previously unavailable tool and initial results.

2) I therefore think the authors have two choices. One alternative is that they should shorten the paper and just present it as a proof of concept. The other is that they should add to it – possibly involving new analyses but certainly new data treatments, to try to give new insights into what benefits such a technique might bring.

In keeping with our reply to point 1) we have introduced a new subsection with figures and related text that display our 2D mapping of elemental concentration at specific sample cross-sections. This contribution resulted from several parallel tracks run parallel to the main ablation tracks. It helps to clarify the spatial variability of element concentrations at the (sub)mm-scale, and therefore allows further discussions about mobility of elements and soluble/insoluble impurities.

3) The positive part is that the authors have successfully used laser ablation to determine 5 elements at 200  $\mu$ m resolution. They describe the way they cleaned the samples (partly with the laser) and the novel way in which they produced quasi-homogeneous standards. I congratulate them on this.

**Thank you.**

4) The headline findings from the study are not new: that dust elements change very rapidly

(annual scale) at the start of a D-O event (this was already said as far back as Fuhrer et al 1999), and that they appear to change before the water isotopes (already covered by Steffensen et al and Thomas et al). Sure, this is the oldest section on which such a finding has been confirmed, but as only one event is studied it just adds an example rather than offering a generalisation, and certainly doesn't provide evidence to make new ideas about the mechanism. Of course, this is not the authors' fault. On the other hand they could have taken the opportunity to really discuss what the advantages and drawbacks of such high resolution might be. I can suggest several lines of study they could have taken:

We consider our contribution relevant not only methodologically but also application-wise as we show – as R1 (and indeed also R2) state and acknowledge as novelties – a way to achieving sub-annually resolved data for one of the very early Greenland DO-events (or indeed for other low accumulation sites). Yes, only one, but at least one of the very early ones. We simply do not consider this manuscript space-wise to be the appropriate place to show more DOtransition data plus an extended discussion; this is planned for another contribution about to be submitted for publication.

5)

An obvious issue is how reproducible the data from such narrow tracks are. The authors say they ran parallel tracks but then do not show us the data so we can assess. I don't know how far apart the tracks were, but parallel tracks across the core at cm distances would have given a crucial clue to reproducibility, which in turn would allow a conclusion as to whether the advantages of high resolution are real (providing evidence of climate variability) or illusory (providing evidence of depositional noise).
We have added a paragraph in the methodology section addressing track

we have added a paragraph in the methodology section addressing track reproducibility, illustrated by a figure in the supplementary material (new Fig. S3) that displays two parallel tracks (2 mm apart) along three consecutive samples for a total of 15 cm. The data show that the patterns generally preserve the overall shape and the intensities maintain similar absolute values over the entire length, with local variations induced by a differential presence on the ablation track of micro-particles (possibly) or grain boundaries and triple junctions.

2. A second issue concerns diffusion. It is generally assumed that water isotopes diffuse a few cm in the firn and then also in solid ice, sulfate peaks appear to diffuse, while dust probably does not diffuse. What about these elements? Here are data apparently showing the retention of mm scale structure at 80 ka ago. This is interesting in its own right and would be even more so if compared to the structure at the start of DO events in the younger part of the record. It might even have been possible to derive diffusion coefficients, which might be crucial when investigating even older ice (eg in Antarctica). Our 2D maps outline a pattern of elevated elemental concentrations in the proximity of grain boundaries (but see also Della Lunga et al., 2014). Furthermore, 2D maps of the most soluble ('sea-salt') proxies show a closer match between the high concentrations zones and the grain boundary network. This could be ascribable to a relative difference in the source of the elemental signals, being increasingly related to randomly dispersed dust micro-particles going from Na to Fe. A quantitative treatment of diffusion coefficients goes beyond what we had intended to present in this contribution but is certainly something to be reported in a future publication as it

**is contained in the PhD thesis of the first author already (DDL).**

3. What is this method actually analysing and how does that compare to what CFA and IC measure? We are shown a comparison only for Na (not counting dust which cannot be compared quantitatively). Why? This seems crucial and even if the data are not yet available from the CFA for eg Ca (which is odd if Na has been measured), it would have been trivial to prepare a few 1 cm samples for IC analysis. This seems critical because Fig 5 seems to show unexpectedly poor agreement for Na, which certainly needs discussion. But in general the consideration of whether this method measures more of the insoluble component than CFA/IC would have been an important analytical discussion that could have been included.

Following on from our previous comment, we introduced further discussions of Na data and especially the differences between CFA and LA-ICPMS in terms of soluble/insoluble particles analysed. Unfortunately, high resolution Ca (CFA data) is not available for the corresponding depth interval. However, we do want to stress that this information requested by R1 is in part already contained in the original manuscript in form of Fig. S4 (now, former S3), which shows a comparison of cryo-cell-LA and solution ICPMS data.

I will discuss a few details below, but as already outlined, the issues above could be discussed; if the authors prefer not to then the paper should be cut back to an analytical proof of concept.

Detailed comments:

6) Page 2, para 1. You seem to come down on one side of an ongoing discussion about whether the cold period enhancement is mainly due to increased transport or to the presence of a sea ice source. It would better reflect the science if you left that open.

We have added a sentence clarifying that the role of salty brines or blowing snow on top of sea ice it is still a matter of debate concerning their contribution to the wintertime peak in sea salt aerosol.

7) Page 2 line 23. It gives a misleading impression to state that rge resolution is "nominally…weekly" because precipitation intermittency and snowdrift mean that weekly resolution is certainly not available. I suspect you know that with your use of the word "nominally", and you should explain that.

**We changed the expression to "50 data points per year", avoiding the misleading impression.**

8) Page 3, line 1. You say that the section "covers" GI21.2, and then give an age range of 370 year (84.70-85.07 ka) for that. But in the abstract you refer to it as a 250 year section, even though Fig 1 shows that it is actually wider than GI21.2. This is incompatible – please correct. The ages of GI21.2 have now been corrected and clarified. Sorry for the confusion and thanks for pointing out the inconsistency.

9) Page 3, line 16. Sorry to be picky but you cite Fig 5 before Figs 2-4. This sentence has been moved towards the end of the methodology section.

10) Eq 1 and line 25 is confusing. If I understand it m\_i is the slope of intensity vs time, whereas your wording made me think it was the slope of the calibration (intensity vs standard concentration). Please clarify. I think Fig 1 would be better shown as linear rather than log plots, as the log plot hides the extent of the drift.

We have added in brackets the reference to Fig S1 to clarify that the mstd\_i coefficient refers to the slope of instrumental drift. This figure was changed and now shows a linear y-axis.

11) Page 4, line 25. Is this the R2 of lin-lin or log-log plots? You show log plots but then describe it as a linear regression. Please clarify.

R2 and linear regression slopes were calculated from lin-lin plots. The plot utilizes log-log axis for the sake of display only. It has been now clarified in the text how the R2 values and slope coefficient were derived.

12) Page 4, line 31. Here is where you say you analysed two parallel tracks to assess reproducibility but then you never do so.

New Fig S3 now displays two parallel tracks on three consecutive samples to assess reproducibility. A paragraph in the text was added to illustrate the figure.

13) Results, page 5-6, seems repetitive (last para page 5 and first para page 6). Combine them into something clearer?

The Results section has been revised; we removed some of the repetition in the first part and added a paragraph towards the end to describe the figures added after this revision.

14) Page 6, line 9, should be Figs 6 and 7 not 8 and 9. Noted and changed.

15) Page 6, data comparison, lines 27-32. It is clearly not true that Na is comparable between the two techniques. While they match OK at 2689.7-2690.0, they are at least a factor 3 off in the shallower section. This needs a better and more correct discussion. (And of course I would like to see the same for Ca).

Discussion concerning Na data and soluble/insoluble origin of 'sea-salt' and 'dust' proxies has been added to address some of the discrepancy between CFA and LA-data and what can be concluded from that.

**Methods and Calibration**

This section corresponds to more than two hundred years, given the observed layer thickness of ~10 mm (Vallelonga et al., 2012). In the flow-model-based GICC05modelext timescale, the section covers an age range of 85.09 - 84.86 ka b2k and includes the 100-year long GI-21.2 and the transitions in and out of this period (Wolff et al., 2010; 
[revised manuscript text omitted]

---

## Author Comment (AC2) · 2 Mar 2017

**The Authors would like to thank R2 for the helpful comments that helped to revise and improve the manuscript. Our replies are here in bold. Relevant changes in text and figures are illustrated at the end of this document, where added or modified text is highlighted in yellow.**

1) This is an interesting MS showing the powerful application of LA-ICPMS for high resolution (200 um) ice core analysis. This may have a lot of implication especially for low accumulation sites and/or abrupt changes.

**We are glad to see that R2 agrees with us that our cryo-cell LA-ICPMS methodology not only is powerful but also widely applicable.**

2) There are some novelties in this paper on the way the authors prepared the standards to convert the count per seconds (intensities) into concentrations. However they use standard riverine waters (SLRS) and a suspension of NIST648 leached with ultrapure HNO3, which resulted in an ice matrix standard which is far from the real ice matrix. We know that the slopes of the calibration curves are highly dependent from the matrix itself and I think therefore that the results may be strongly biased by the different ionization of the ice matrix compared to the standard ones.

**Ultimately we are ablating the same $H_2O$ matrix in both samples and standards, so we respectfully disagree with R2 that they are 'far from the real ice matrix'. Using such artificial ice standards to calibrate LA-ICPMS analyses of ice goes back – with contrasting results as to the resultant homogeneity - to the pioneering work of Reinhard et al (2003), or Sneed et al. (2015).  In fact, 193 nm excimer laser-ablation is known to be relatively more matrix-tolerant (Guillong et al., 2003) compared to other ns-LA methodology) such that, for example, *accurate* results can be obtained for as contrasting matrixes such as carbonate samples standardized with Na-Ca-Al-Si-glasses (the NIST61x glass suite). If available, we would appreciate a reference supporting the assertion 'We know that the slopes … are highly dependent from the matrix itself…', to be able to evaluate this ourselves.**

**However, we want to stress that our standardization is far from being perfect given the problem of making very homogenous ice stds. In the absence of a usable internal std (OH-signal is not sufficiently above background), this level of matrix-matching is actually state-of-the-art and thus the currently best available.**

3) The authors should comment on this and clearly demonstrate how they obtain real concentration and not just relative changes that can be easily seen just looking at the variation of intensities.

**All of our Ice standards were prepared by dilution between 1:10 and 1:1000 of the certified reference material with ultrapure $H_2O$ (>18 MΩ·cm); we very mildly acidified these solutions with 1% ultrapure $HNO_3$ to stabilize them before freezing and to align them with the acidity of the multi-elemental standard solution ICP1 (Sigma-Aldrich), which was the only one being originally (before dilution) in 10% $HNO_3$, unlike all of our other standard solutions. We believe any bias introduced is contained within the ~ 16-**

**18 % error involved in the calibration procedure. This has been now clarified in the methodology section.**

4) In addition, the authors claim to use 24Mg, 27Al, 40Ca, 56Fe. All these masses are highly interfered by spectral and matrix interferences in ICPMS. Despite I think they used a SF-ICP-MS or a collision-cell instrument to reduce the interferences, I think that a better description of the methodology should be given. I know that most of the details are given in Della Lunga (2014), but a minimal description of the methodology is compulsory.

**We have added several details regarding the methodology, including a description of the removal of interferences by the use of $H_2$ in the Agilent 7500cs reaction cell; but we also note than these details are provided in our earlier publication in the Journal of Glaciology which had been included at the outset (Della Lunga et al., 2014). If the ice matrix was contributing to the signals on m/z=24, 27, 40, 56 then we would not obtain no resolvable counts from instrumental background for our ice blanks.**

5) Then, I do not fully understand the objective of the MS, since most of the findings are not new at all. I would have rather focussed the MS into a comparison between LA-ICPMS vs CFA, but this would have required a more robust statistical tool.

I would therefore suggest the authors to readdress the MS to a specific target: i) analytical (in this case the paper lacks of many details), describing in detail the new advancement of this powerful technique and duly comparing the data with CFA results; in this case the reproducibility of the analysis on different sections is a key parameter, but as far as I can see there are no evidence of this in the paper; ii) more oriented toward a climatic/environmental interpretation; in this case the real benefit of the LA-ICP-MS approach should have been better explained.

**As we also state in the reply to R1, the focus of the manuscript was predominantly methodological yet with one integral case study of deep ice to illustrate the power of the significantly improved, in fact unprecedented, spatial resolution. To make this focus clearer, the revised manuscript now presents a stronger methodological section and a more detailed description of the signal emerging from ablation with 193 nm laser and its possible interpretation. The climatic picture emerging from our analysis of GI21.2 is intended to demonstrate the capability of the technique to recover extremely small scale variability together with Stadial-Interstadial fingerprint in dust and sea salt proxies for one of the most abrupt and short lived transitions in deep ice where ice layer thickness is becoming very small.**

**Rerefences**

Della Lunga, D., Müller, W., Rasmussen, S. O., & Svensson, A. (2014). Location of cation impurities in NGRIP deep ice revealed by cryo-cell UV-laser-ablation ICPMS. Journal of Glaciology, 60(223), 970-988.

Guillong, M., Horn, I., & Günther, D. (2003). A comparison of 266 nm, 213 nm and 193 nm produced from a single solid state Nd: YAG laser for laser ablation ICP-MS. Journal of analytical atomic spectrometry, 18(10), 1224-1230.

Reinhardt, H., Kriews, M., Miller, H., Lüdke, C., Hoffmann, E., & Skole, J. (2003). Application of LA–ICP–MS in polar ice core studies. Analytical and bioanalytical chemistry, 375(8), 1265-1275.

Sneed, S. B., Mayewski, P. A., Sayre, W. G., Handley, M. J., Kurbatov, A. V., Taylor, K. C. & Spaulding, N. E. (2015). New LA-ICP-MS cryocell and calibration technique for sub-millimeter analysis of ice cores. Journal of glaciology, 61(226), 233-242.

**Post Review changes to:**

**Calibrated cryo-cell UV-LA-ICPMS elemental concentrations from NGRIP ice core reveal abrupt, sub-annual variability in dust across the interstadial period GI-21.2**

Damiano Della Lunga[1], Wolfgang Müller[1], Sune Olander Rasmussen[2], Anders Svensson[2], Paul Vallelonga[2]

[1]Department of Earth Sciences, Royal Holloway University of London, Egham TW20 0EX, United Kingdom
[2]Centre for Ice and Climate, Niels Bohr Institute, University of Copenhagen, 2100 Copenhagen Ø, Denmark

**Introduction**

[……] This mechanism, which is thought to be the primary reason for sea-salt enrichment in ice cores during cooling events, receives further contributions of sea salt from another source. When sea ice is formed, highly saline brine and fragile frost flowers form on top of the frozen surface. This brine represents a further source of aerosol, carried over land by the wind (Wolff et al, 2003). However, a quantitative assessment of the contribution of brine, frost flowers, and blowing snow to the wintertime peak in sea salt aerosol it is still a matter of debate (Huang and Jaegle, 2016).

[……]The aim of the present study is to assess the sensitivity and the phasing of dust/sea-salt proxies as $Na^+$, $Fe^{2+}$, $Al^{3+}$, $Ca^{2+}$ and $Mg^{2+}$ at a resolution of ~200 μm (providing approximately 50 data points per calendar year at this depth) across the abrupt warming into and cooling out of the precursor event GI-21.2. Furthermore, we present an updated fully quantitative calibration for the elements under investigation, following Della Lunga et al. (2014) and Müller et al. (2011).

**Methods and Calibration**

This section corresponds to more than two hundred years, given the observed layer thickness of ~10 mm (Vallelonga et al., 2012). In the flow-model-based GICC05modelext timescale, the section covers an age range of 85.09 – 84.86 ka b2k and includes the 100-year long GI-21.2 and the transitions in and out of this period (Wolff et al., 2010; 
[revised manuscript text omitted]

[Figure]

**Figure S3: Reproducibility of LA tracks: example of raw intensities (cps) of main elements from 2 mm apart parallel ablation tracks acquired over 15 cm (3 samples) between depths of 2691.45-2691.30 m (left to right).**

[Figure]

**Figure S4: Comparison between solution data and cryo-cell UV-LA-ICPMS data on three different samples corresponding to three different 50-mm depth intervals: 4940A11 (depth range: 2716.45 – 2716.50 m), 4900A3 (depth range: 2694.85 – 2694.90 m) and 4882B4 (depth range: 2684.875 – 2684.925 m), representing small sections of early GS-22, late GS-22 and GI-21.1, respectively. Results show that LA and solution data differ of only 5 – 20% and therefore are within error margins. See text for details.**

**Table S1: Concentration of elements under investigation in aqueous reference materials used for ice standard preparation: SLRS-5-"River water reference material for trace metals" (National Research Council of Canada, diluted 10 times [SLRS-5_10] when not specified), Water low (RHUL internal standard), 90243 Multi-element standard solution 1 for ICP (Sigma Aldrich, diluted 20 times), and NIST SRM 1648 Urban Particulate reference material (in suspension, see text for details). Blank concentrations (in ppb) of ultrapure water at RHUL were obtained in solution mode and are shown on the right column. Limit of detection (LOD) refer to cryo-cell LA-ICPMS analyses only.**

| | Standard name | | | | | |
|---|---|---|---|---|---|---|
| | SLRS-5_10 | ICP-20 | Water Low | NIST1648a | | RHUL Deionized water |
| Element | concentration (ppb) | concentration (ppb) | concentration (ppb) | concentration (ppb) | LODs LA-ICPMS (ppb) | (ppb) |
| Al | 4.95 ± 0.5 49.5 ± 4.8 (SLRS-5) | 2525±2.5 | 9.8±0.1 | 1683±16 | 1.12 | 0.012 |
| Fe | 9.1 ± 0.6 91 ±6 (SLRS-5) | 505±0.5 | 9.8±0.1 | 1924±20 | 1.06 | 0.627 |
| Ca | 1050 ± 40 10510±380 (SLRS-5) | 505±0.5 | 48.9±0.3 | 3126±15 | 0.63 | 0.606 |
| Mg | 254 ± 16 2541±155 (SLRS-5) | 505±0.5 | 48.9±0.3 | 394±5 | 0.92 | 0.021 |
| Na | 538 ± 10 5380±105 (SLRS-5) | 2525±2.5 | 244±1.5 | 209±4 | 48.3 | - |

---

## Author Response (AR2)

**Authors reply to Reviewer 1 – 2nd iteration**

**The Authors are glad that the additions and modifications to the manuscript have been welcomed and would like to thank R1 for the helpful and supportive comments. Original review comments are here followed by our replies in bold.**

The authors have improved the paper and answered many of the points I raised in my original review. I am particularly pleased that they have added an interesting section showing the benefits of their technique for looking at microstructure (Figs 9 and 10). There are also improvements in the description of the method and in the interpretation of the data.

While the paper is still quite light on application, it is now suitable for publication after moderate revision. There are still a few issues that need addressing, mainly typos, but at least one remaining science issue to be resolved:

Page 2, line 7 "from a quantitative", needed to make the sentence make sense.

**The sentence has been corrected.**

Page 4 line 9 "matrix is a satisfactory"

**Noted and changed.**

Page 4, line 34 "shows all the regression"

**Noted and changed.**

Page 5, line 8-10. I don't find Fig S3 very satisfactory because it is difficult to see how similar the two tracks are. It would be much better to plot the two tracks for each element together in 5 panels, 1 per element, if you want to illustrate this point. In fact I am not convinced the description "identical absolute values" is correct. Taking Ca as an example, at 12-15 cm, one track shows 10^4-10^5, while the other shows 10^5-10^7. I see large differences for Al at 0-1 cm too. Please give a more careful description of inter track variability.

**The figure and the corresponding text have been modified as suggested.**

Page 5, line 11 "resulting" not "resulted".

**Noted and changed.**

Page 5. Please be really clear: you used two parallel tracks and averaged in this example. Was that done for all the plots shown or are they single tracks?

**All the profiles shown in the manuscript result from averaging at least 2 parallel original laser scan along the samples, except for figure S3 where two single tracks are compared. In the case of laser spots the data result from a single laser drilling. This has been clarified in the text.**

Page 6. I may have missed it but I don't see a call to Fig 8 which actually seems unnecessary.

**Figure 8 is mentioned at page 6, line 23.**

Page 7. ESSENTIAL EDIT: While the comparisons in Fig S4 are not too bad, they are few samples (and considerably off for 4882). But the comparison of Na in Fig 5 is very poor and cannot be rescued. As I said before, there are sections where LA-ICPMS and CFA give similar values and sections where they differ by a factor 5. This plot simply shows that there is a problem for Na, something you already admitted. It should either be removed, or used to confirm that LAICPMS cannot be used yet for Na. It cannot be used at all to bolster any argument that the two techniques agree.

**This part has been partially revised indicating all the problematics related to the interpretation of Na in this study.**

Page 9, lines 18-24. This can really not be the reason for a factor 5 offset of LA-ICPMS Na. The issues you raise would lead to short range variability about an identical mean value. For Na you have a completely different mean in places. Please edit.

**This part has been also revised accordingly. See previous comment.**

Page 9, line 27. "concentrations of impurities is" (remove "it").

**Noted and changed.**

Page 10, line 3. So did you use two tracks in this study? It's really not clear, and seems unlikely as this would have smoothed some of the variability.

**The 2D maps were constructed via static laser drilling as described at page 5, line7-9.**

Fig 1 still has a problem as the box should run to 2691.5 m but actually goes to 2691.1 m.

**The figure has been modified.**

[revised manuscript text omitted]